# Orthorectification of Helicopter-Borne High Resolution Experimental Burn Observation from Infra Red Handheld Imagers

Ronan Paugam [1,*], Martin J. Wooster [2,3], William E. Mell [4], Mélanie C. Rochoux [5], Jean-Baptiste Filippi [6], Gernot Rücker [7], Olaf Frauenberger [8], Eckehard Lorenz [9], Wilfrid Schroeder [10] and Bruce Main [2] and Navashni Govender [11,12]



1 Centre for Technological Risk Studies, Department of Chemical Engineering, Universitat Politécnica de Catalunya—BarcelonaTech, Diagonal 647, E-08028 Barcelona, Spain
2 Environmental Monitoring and Modelling Research Group, Department of Geography, King's College London, London WC2R 2LS, UK; martin.wooster@kcl.ac.uk (M.J.W.); bruce.main@kcl.ac.uk (B.M.)
3 Leverhulme Centre for Wildfires, Environment and Society, Department of Geography, King's College London, London WC2B 4BG, UK
4 Wildland Fire Science Lab, USDA Forest Service, 400 N 34th St., Suite 201, Seattle, WA 98103, USA; william.mell@usda.gov
5 CECI, Université de Toulouse, 31100 Toulouse, France; rochoux@cerfacs.fr
6 Laboratory for the Physical Systems of the Environment, Université de Corse, 20250 Corte, France; filippi@univ-corse.fr
7 ZEBRIS Geo-IT GmbH, Lipowskystr 26, D-81373 Munich, Germany; gruecker@zebris.com
8 German Remote Sensing Data Center, German Aerospace Center, Kalkhorstweg 53, D-17235 Neustrelitz, Germany; olaf.frauenberger@dlr.de
9 Optical Information Systems, German Aerospace Center, Rutherfordstraße 2, D-12489 Berlin, Germany; eckehardt.lorenz@dlr.de
10 Satellite Analysis Branch, NOAA, NESDIS, College Park, MD 20740, USA; wilfrid.schroeder@noaa.gov
11 Conservation Department, Kruger National Park, South African National Parks, Private Bag X402, Skukuza 1350, South Africa; navashni.govender@sanparks.org
12 School of Natural Resource Management, Nelson Mandela University, Private Bag X6531, George 6530, South Africa
* Correspondence: ronan.paugam@pm.me

**Abstract:** To pursue the development and validation of coupled fire-atmosphere models, the wildland fire modeling community needs validation data sets with scenarios where fire-induced winds influence fire front behavior, and with high temporal and spatial resolution. Helicopter-borne infrared thermal cameras have the potential to monitor landscape-scale wildland fires at a high resolution during experimental burns. To extract valuable information from those observations, three-step image processing is required: (*a*) Orthorectification to warp raw images on a fixed coordinate system grid, (*b*) segmentation to delineate the fire front location out of the orthorectified images, and (*c*) computation of fire behavior metrics such as the rate of spread from the time-evolving fire front location. This work is dedicated to the first orthorectification step, and presents a series of algorithms that are designed to process handheld helicopter-borne thermal images collected during savannah experimental burns. The novelty in the approach lies on its recursive design, which does not require the presence of fixed ground control points, hence relaxing the constraint on field of view coverage and helping the acquisition of high-frequency observations. For four burns ranging from four to eight hectares, long-wave and mid infra red images were collected at 1 and 3 Hz, respectively, and orthorectified at a high spatial resolution (<1 m) with an absolute accuracy estimated to be lower than 4 m. Subsequent computation of fire radiative power is discussed with comparison to concurrent space-borne measurements.

**Keywords:** fire behavior; experimental burn; image processing; orthorectification; infra red



## 1. Introduction

High-resolution active fire monitoring and an associated fire behavior metric are growing needs in the fire science community, in particular in the development of coupled fire-atmosphere systems [1]. To support these efforts, we present here a new approach to process Infra Red (IR) observation from landscape-scale (>100 m) experimental fires, specifically helicopter-borne images collected by handheld Long Wave Infra Red (LWIR) and Middle Infra Red (MIR) cameras. This monitoring system enables far more spatial and temporal details to be extracted than from traditional overpasses by a fixed-wing survey airplane.

## 2. Background

The global wildfire activity shows sign of a decreasing trend over the last two decades [2]. While its health and societal impact was recently estimated on the basis of fine particle matter emissions [3], its associated environmental impact is still difficult to estimate as limited global data sets are available, and its economical impact lack data to be quantitative [4]. However, one clear trend is that in fire-prone regions such as western North-America or south-eastern Australia, wildfire activity has increased over the same time period [2,5] because of severe drought resulting from climate change [6] and increase of human density in Wildand-Urban Interface (WUI) [4]. To improve mitigation of wildfire effects in fire-prone regions, fire-atmosphere coupled systems have been developed and are now intended to become operational [7]. These coupled systems include CAWFE [8], WRF-SFIRE [9,10], and MesoNH-ForeFire [11,12].

To pursue this model validation and development effort, we need data sets that simultaneously monitor fire front (heat release rate, geometry, propagation), plume dynamics (convective flux, geometry), and atmospheric state (ambient profile), in ideally ($a$) a complex scenario where fire-atmosphere interactions influence the fire front dynamics, and ($b$) a timely manner that can capture the coupled system dynamics [13]. Several field-work campaigns have been designed with such intention, but unfortunately they came out with limited data on fire behavior monitoring. In the RxCADRE experiment [14], burns were set in heterogeneous vegetation and with complex ignition patterns that make them difficult to monitor without fast-return observations. The return time of the fixed-wing airplane that was operating the thermal camera could be as long as 1 min [15]. According to Rate Of Spread (ROS) ground measurement made on site [16] (e.g., $ROS^{max} = 0.44 \, \text{m s}^{-1}$), the fire can spread by up to 25 m during that time interval, while a maximum fire front depth of 3 m was recorded [16]. This means that between two observations, a fire front could spread a distance almost 10 times larger than its depth. If the fire was propagating through heterogeneous vegetation, then most certainly at the new observation time the fire would be completely different from what it was at the previous observation. The requirement of high-frequency airborne observations is also reported in the detailed analysis of [17] from a pine stand experimental burn campaign. More recently, in the FireFlux II experiment corresponding to a large homogeneous grass fire [18], thermal images were used to monitor the fire front but from a low vantage point. This made flame distortion effects too important to accurately estimate fire front geometry and fire activity over the whole fire duration. The RxCADRE and FireFlux II examples highlight the difficulty to acquire informative data for the different components of the coupled fire-atmosphere models.

Landscape-scale fire behavior monitoring is however achievable with an IR handheld thermal imager operated from a hovering helicopter. Such a platform makes it possible to acquire high-frequency observations from a near-nadir view point at a relatively low cost as compared to survey aircraft. Both LWIR [19,20] and MIR [21] imagers have been used to collect IR images from burns of several hectares at a high spatial (>1 m) and temporal (>1 s) resolution. Such images were used to compute comprehensive measures of fire behavior metrics, e.g., ROS [19,21], Fire Intensity (FI) [19], and fire radiative heat flux (Fire Radiative Power, FRP, [21]). Hovering Unmanned Aerial Vehicle (UAV) platforms are also quite promising for prescribed/experimental burn application. However, their use for

active burn monitoring is still restricted to small burns [22–24] (10 m), making present UAV applications to landscape-scale monitoring difficult.

To extract fire metrics from the collected IR observations, image processing tasks are required and can be divided into three steps: (*a*) orthorectification, which consists in warping the raw images on a fixed coordinate system grid to correct for camera lens distortion and perspective effects induced by camera orientation and terrain; (*b*) segmentation, which involves delineating the fire front location out of each Orthorectified image; and finally (*c*) fire behavior metrics computation (e.g., ROS) from consecutive fire front locations. One main difficulty when dealing with images that were collected with a handheld imager is that cameras are usually not coupled with an inertial measurement unit as in survey aircraft [15]. Therefore, images contain no information on the camera position and orientation (hereafter named camera pose), making their orthorectification challenging. For example, in the work of [19,25,26], each image is orthorectified using a manually-selected pixel location of a known geographic position (i.e., Ground Control Point, GCP). At least, four GCPs are required per image [27]. GCPs are commonly set with fires lighted at the corners of the burn plot [19,21,25]. Hereafter, these GCPs are named corner fires. The "corner fire" approach comes with several constraints. Every image for which less than four corner fires are visible or identifiable would have to be disregarded. This makes high-frequency observation acquisition more challenging because controlling a camera operated from an hovering helicopter can be difficult when the aircraft is subject to atmospheric turbulence, potentially enhanced by the fire. This also implies that the burn plot needs to be set with hot corner fires prior to fire ignition, imposing management constraints on the fire crew.

The development of a methodology to perform orthorectification (step *a*) to a large number of IR fire images collected from a moving platform and at a high frequency is a current need of the fire science community [23]. A recent work [28] proposed a first attempt to stabilize (image-to-image registration) a time series of IR images. However, the proposed methodology has some limitations since it only features good performance for images recorded from a stable vantage point (see drift in all footage of the Supplementary Material of [28]).

## 3. Objectives

This work is part of an effort to simplify IR monitoring during landscape-scale experimental burns, with the objective to compute fire behavior metrics from MIR and LWIR observations collected with a high frequency imager delivered without information on camera location and orientation. In continuity with the work of [21], we design and evaluate a method to orthorectify landscape-scale experimental burn observations. The constraint on corner fires is removed (there is no need of fixed GCPs present during the whole fire duration). However, the obligation of having a set scene remains (the final burnt area needs to be delimited), as well as a burning plot with a constant slope terrain.

This article is structured as follows. Section 4 presents the available data. Section 5 gives an overview of the orthorectification algorithms and their limitations. Section 6 shows the algorithm results. Finally, Section 7 discusses orthorectification accuracy as well as an application to compute FRP and its time integration, the Fire Radiative Energy (FRE). Supplementary material provides a detailed description of the image processing algorithms. Tables, images and equations from the supplementary material are referenced with a S (e.g., Figure S4).

## 4. Experimental Burn Data

Data were collected in late August 2014 during a series of four experimental burns conducted in Kruger National Park (KNP, South Africa) at a time around the peak of the region's fire season (Table 1).

**Table 1.** Summary of the four KNP14 experimental burns. Burns are named after the strings of plot it belongs and its number in the strings. There are 16 strings in KNP, which account for as many replicates used in the experimental burn plot trial initiated in 1954 [29]. The table entry "$\sigma$ terrain elevation" corresponds to the standard deviation of the plane surface approximation difference with the terrain model from the NASA Shuttle Radar Topography Mission [30]. All plots are shown here at the same scale.

| Burn Plot | Skukuza4 | Skukuza6 | Shabeni1 | Shabeni3 |
|---|---|---|---|---|
| First available visible image. White arrow shows the North. |  |  |  |  |
| Ignition day | 26 August 2014 | 26 August 2014 | 22 August 2014 | 22 August 2014 |
| Ignition time (LT) | 13:26 | 10:59 | 13:00 | 11:00 |
| Plot size (ha) | 4.3 | 7.0 | 8.2 | 7.4 |
| $\sigma$ terrain elevation (m) | 4.8 | 1.12 | 4.3 | 3.3 |
| Fuel load (kg/ha) / moisture (%) | 4128/15.4 | 2654/22.8 | 4777/25 | 4678/16.9 |
| Average $T$ (C) | 33 | 30 | 32 | 28 |
| Average $RH$ (%) | 48 | 60 | 23 | 42 |
| Mean wind speed (m s$^{-1}$) | 1.42 | 2.63 | 0.45 | 2.71 |
| Mean wind direction (°) | 140 | 160 | 320 | 320 |
| Fire duration (min) | 9.3 | 17.7 | 20.0 | 25.2 |
| Corner fire | yes | yes | no | no |
| Number of images LWIR/MIR/VIS | 486/1437/396 | 980/1838/1065 | 527/1819/650 | 1650/1513/1620 |
| Matching satellite | VIIRS | TET | VIIRS | TET |
| Comments | Smoky plume and large front depth. | Multiple fronts. Weak fire on the eastern side. MIR camera set with a filter blocking 30% of the radiation. | Intense fire, with two spotting ignitions outside the plot. | Slow moving backfires followed by one intense front merging. LWIR images are slightly blurred. |

Each fire is conducted in one of the long-term experimental burn plots (<8 ha) covered with savannah-type vegetation [31]. Plots are well approximated with a plane surface Digital Elevation Model (DEM) (see standard deviation of plane surface approximation to DEM from the NASA Shuttle Radar Topography Mission [30], "$\sigma$ terrain elevation" in Table 1). Following the KNP nomenclature plot name, the four fires are named: Shabeni1, Shabeni3, Skukuza4, and Skukuza6.

Fuel load and moisture are estimated with pre-fire in situ destructive sampling (Table 1). Whilst there were many standing trees on each plot (e.g., Combretum, Sclerocarya birrea, and Terminalia sericea), the fire appeared to leave them most unaffected. Grasses (which can exceed 1 m height) are the bulk of the consumed fuel. Further plot descriptions related to previous KNP field campaigns can be found in [32], who focused on pre-/post-fire reflectance simulations, or in [33], who developed an emission factor measurement method.

Each burn plot is first ignited with a backfire to create a fire break at the downwind side (see backfire examples in Table 1 for Skukuza4 and Shabeni1). Headfire ignition follows some minutes later at the upwind side of the plot. During each burn, radiative energy emissions from the burning fuel were assessed using two IR thermal imaging cameras, one operated in LWIR (Optris PI 400) and the other one in MIR (FLIR Agema 550). A third camera operating in the visible (VIS, Gopro Hero 2 with no IR filter) was set on the same portable mount than the IR cameras. It is used for qualitative purposes only to link IR fire activity observation to basic observer features (flame location, plume development), and its image orthorectification are then not discussed in this paper. Figure 1 shows an example of concurrent VIS, LWIR, and MIR camera observations around the peak activity of the Shabeni1 burn.

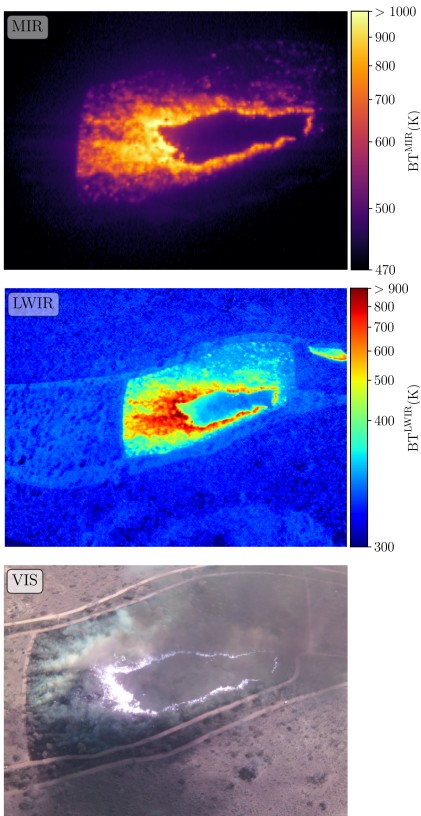

**Figure 1.** Example of raw images captured by the three cameras operated during the KNP14 field campaign: visible (VIS), Long Wave Infra Red (LWIR) and Middle Infra Red (MIR). The concurrent images were collected at $t = 951$ s after ignition of the Shabeni1 burn. Cameras specifications are reported in Table 2.

An important difference between the two IR cameras is their operating temperature range (Table 2). While the Optris PI 400 camera can record LWIR Brightness Temperature (BT) from the background temperature (290 K) to a temperature typical of a fire scene (600–1000 K), the Agema 550 camera, when setup to monitor in a temperature range above 600 K, cannot sense MIR background BT $470 < BT^{MIR} < 1073$ K, [21].

The limited plot size (<8 ha) allows for a hovering altitude of 600 m, which makes it possible to orthorectify IR images at a 1-m spatial resolution (Section S1.1). Skukuza4, which is a smaller plot, is monitored at a lower altitude, hence orthorectified at a higher resolution of 50 cm. The data set built up during this field work is named KNP14.

During the burn, the helicopter pilot tried to hover as much as possible at the same location near the plot edge, while the camera operator kept as much as possible the full extent of the burning plot in the MIR camera field of view (the MIR camera has the smallest field of view among the three cameras, Table 2). The resulting data set is three sequences of images from the LWIR, MIR, and VIS cameras, collected at different frame rates (3, 1, and 1 Hz, respectively). The three cameras are not time synchronized; two of them (LWIR and VIS) had an unstable frame rate. Therefore, warping transformations derived for one camera are not directly applicable to the other cameras. Another constraint of the KNP14 data set is the lack of detailed camera calibration. Using a pinhole model, we only have access to prior knowledge of the intrinsic camera matrix, but not the associated distortion coefficient. Implications are discussed in Section 6. Section 5 assumes that the camera calibration is known.

For each burn, the ignition time was chosen to match with satellite overpass, providing fire-dedicated sensor observation and in particular FRP measurements (Table 1). Two sensors were targeted: The Visible Infrared Imaging Radiometer Suite (VIIRS) aboard the joint NASA/NOAA Suomi spacecraft with two active fire products at 375 and 750-m [34], and the experimental TET-1 (Technologie Erprobungs Träger-1) micro-satellite developed by DLR with a 178-m resolution [35]. These two sensors deliver, at the time of the experiment, the highest resolution active fire satellite products. Section 7.3 provides a comparison of FRP computed from space-borne observation against FRP derived from the orthorectified helicopter-borne images.

**Table 2.** Specification of the cameras used to monitor the KNP14 experimental burns. The three cameras were operated simultaneously, handheld on the same mount, from a helicopter hovering above the fire looking at the fire from off-nadir.

| Camera | Optris PI 400 LWIR | Agema 550 MIR | Gopro Hero 2 VIS |
|---|---|---|---|
| Wave length (µm) | 7.5–14 | 3.9 | RGB |
| Sensor size (pixels) | $382 \times 288$ | $320 \times 240$ | $3840 \times 2880$ |
| Field of view (°) | 53 | 40 | 58 |
| Nominal frame rate (Hz) | ∼1–2 | 3 | ∼1–2 |
| Temperature range (K) | 253 to 1173 | 473 to 1073 | - |
| Comment | | | IR filter removed |

## 5. Orthorectification Algorithms

This section presents the suite of algorithms designed to orthorectify the LWIR and MIR images (Section 4). Orthorectification corresponds to the process of taking an image in its original geometry and warping it onto a coordinate system (i.e., a fix grid). This process corrects for distortion due to topography variation and camera lens effects. Our approach comes with two main constrains: The terrain is limited to plane surface and the camera needs to keep a near constant line of sight (see Section S1.1 for more details).

There are three main orthorectification issues when dealing with the KNP14 data set: (*a*) Camera locations are not fixed; (*b*) relative camera poses are unknown because of the unsynchronized camera frame rates; and (*c*) the observed scene is constantly modified

while the fire propagates: Presence of smoke and alteration of vegetation from unburnt to burnt. These issues imply that each image requires its own warping transformation, and that only images of near-time acquisition can be matched, the time lapse depending on the fire dynamics and/or the ability of the helicopter to hover.

The cameras used in this experiment differ by their ability to sense or not the background scene. We discuss separately the algorithms developed for the LWIR camera (with background and fire sensing capability) and for the MIR camera (that can only record $BT^{MIR}$ above 470 K).

### 5.1. Algorithms for Long Wave Infra Red

To orthorectify time-series of near-nadir LWIR images, the following procedure is proposed: ($a$) Orthorectify the first image, $I_0$, on a coordinate system using manually-selected GCPs (this yields the computation of $I_0^{rec}$), and ($b$) recursively align each image $I_p$ using a reference image $I_m$ picked from the set of previously-processed images ($I_0, ..., I_{p-1}$).

With the assumption of a constant slope terrain (plane surface DEM) and a pinhole camera model, the initial orthorectification and the image alignment can be formulated both with linear matrix transformations, i.e., homography matrix (Section S1.1). This matrix completes the warping between planar surfaces, yielding $I_0^{rec} \simeq H_{rec}^0 \cdot I_0$ for the first image orthorectification, and $I_p^m \simeq H_m^p \cdot I_p$ for the $p$th image alignment. At the end, the recursive image alignment (i.e., $H_m^p$) is used to propagate the initial orthorectification projection $H_{rec}^0$ to the $p$th image $I_p$ and thereby compute $I_p^{rec} = H_{rec}^p \cdot I_p$.

The constant change of the fire scene requires to adopt a moving reference image $I_m$. If $I_m$ is changed too frequently, it may introduce unwanted drift resulting from the moving feature stabilization. This is probably the reason why the methodology of [28] is showing drift, as stabilization is performed on consecutive images ($m = p - 1$). Inversely, if $I_m$ is updated too slowly, image differences resulting from the fire evolution or from the view angle changes would degrade the alignment performance. The main challenge is to update the moving reference image $I_m$ while limiting error accumulation that is inevitably caused by slight image misalignment.

During the fire, three different areas can be identified in the fire scene: ($i$) The flaming area located just behind the front where flaming combustion is active; ($ii$) the cooling area located behind the flaming area made of cooling ground and residual smoldering combustion; and finally ($iii$) the plume area that is mostly located ahead of the flaming zone but with overlap on the flaming area and potentially extending to smoldering spots of the cooling area. From an image alignment perspective, the flaming and plume areas are problematic as the front and plume can spread with minor changes in their appearance between consecutive images, creating potential outlier features. Conversely, the cooling area has potential interest since it is composed of fixed features whose aspect and/or temperature change at a lower time rate (typically, several minutes) than the camera frame rate (less than 1 s).

To minimize effects from the plume and front, a two-step approach for the calculation of the projection $H_{rec}^p$ is designed. First, an alignment using a mask over the burning plot is done to focus on the background scene and provide an initial guess of $H_{rec}^p$ (Algorithm 1). Second, the projection $H_p^{rec}$ is optimized using cooling area information (Algorithm 2).

#### 5.1.1. Algorithm 1

Algorithm 1 is summarized above. The calibration of the control parameters is discussed in Section 6.1. In the algorithm description, $P^m$ is the plot mask $P$ projected on $I_m$ perspective. $M_\bullet$ is the mask associated to image $I_\bullet$. It is designed to emphasize steady features, and is hereafter named the steady area mask. $M_\bullet$ masks fire pixels using a conservative fix temperature threshold (420 K), but also undesirable foreground features like the helicopter skid (Section S1.2). $\rho_{ECC}$ is the 2D correlation defined in [36] that measures image similarity (Equation (S3)). SSIM is the Structural Similarity Index Metric [37]. $\mathcal{J}_p$ is the ensemble of images indices at iteration $p$ for which correlation to matching reference

image was greater than the threshold $\rho_{\text{ECC}}^{\text{Align}}$, i.e., image aligned with "good" alignment flag. $\mathcal{J}_p^{\text{ref}}$ is the ensemble of image indices used as the reference image. The rules for updating the reference image $I_m$ (i.e., the first step in the loop of Algorithm 1) is central to the algorithm stability. The trend of $\rho_p$ evolution along the iterations was chosen to trigger the update, hence choosing a new reference image when $\rho_p$ remains below $\rho_{\text{ECC}}^{\text{Align}}$ for four iterations. The new $I_m$ image is then selected within $J_p$ with the two conditions of having a correlation greater than $\rho_{\text{ECC}}^{\text{Ref}}$, and being older than $I_p$ by at least $t_{\text{tail}}$ seconds.

---

**Algorithm 1** Iterative orthorectification using LWIR image background

---

$I_0^{\text{rec}} \backsimeq H_{\text{rec}}^0 \cdot I_0$      ▷ manual orthorectification
**for** $p = 1, N$ **do**
     $I_m \leftarrow \left( \mathcal{J}_p, \rho_{\text{ECC}}^{\text{Ref}}, \rho_{\text{ECC}}^{\text{Alig}}, t_{\text{tail}} \right)$ and $\mathcal{J}_p^{\text{ref}} \leftarrow m$      ▷ reference image update
     $H_m^p \leftarrow (I_p, I_m, \mathcal{J}_p, \mathcal{J}_p^{\text{ref}}, P^m, n_{\text{tail}})$      ▷ feature and area-based alignment, see step 5 in Figure S4

     $I_p^m \backsimeq H_m^p \cdot I_p$      ▷ projection on $I_m$
     $\rho_p = \rho_{\text{ECC}}(I_p^m, I_m, M_p^m, M_m + P^m)$      ▷ algorithm stability
     $I_p^{\text{rec}} \backsimeq H_{\text{rec}}^p \cdot I_p \backsimeq H_{\text{rec}}^0 \cdot H_0^m \cdot H_m^p \cdot I_p$      ▷ orthorectification
     $\text{SSIM}^{\text{prev}} = P_{80\%}\left( \text{SSIM}(I_p, I_{p-1}) \right)$      ▷ performance assessment
     **if** $\rho_p > \rho_{\text{ECC}}^{\text{Align}} : \mathcal{J}_p \leftarrow p$      ▷ quality flag

---

Following the work by [38,39], the homography matrix $H_m^p$ is built upon the combination of a feature-based method the pyramidal implementation of the (Lucas–Kanade feature tracker, PyLkOpt, [40]), which uses the $n_{\text{tail}}$ last processed images as a template and a multi-resolution area-based (Enhanced Correlation Coefficient, ECC, [36]) method using the reference image as a template. See Section S1.2 for more details on these two methods.

According to [41], the Normalized Mutual Information (NMI) and 2D correlation are good image similarity metrics for a registration quality assessment of LWIR images in the context of fire airborne observations. In this work, since the 2D correlation is inherent to the area-based alignment method ECC (see Equation (S5)), it was selected to monitor algorithm stability.

One additional image similarity metric, SSIM, is introduced to assess orthorectification quality independently from 2D correlation. Note that SSIM application to fire observation has not been yet discussed in the literature [41]. SSIM is chosen here for its capability to estimate local matching between images according to a combination of intensity, contrast, and structure in a window around each pixel [37]. SSIM, when compared to 2D correlation, has the advantage of not being biased by maximum-to-maximum alignment. SSIM is only considered on orthorectified images, so that every pixel has the same weight when considering statistics per image. The metric used to assess the quality of image orthorectification $H_{\text{rec}}^p$ is thus defined as $\text{SSIM}_p^{\text{prev}} = P_{80\%}\left( \text{SSIM}(I_p, I_{p-1}) \right)$, where $P_{80\%}$ is the 80%-percentile and $\text{SSIM}(\cdot, \cdot)$ is the similarity map that uses a neighbor window 20 m in size to include the vegetation structure (e.g., tree, bushes). SSIM is used for the result analysis in Section 6.

### 5.1.2. Algorithm 2

The second optimization step requires the existence of a dynamical mask, which removes the plume and the front from the alignment procedure. The mask is composed of a fire front delimitation that masks out the flaming and plume, and of a filter that emphasizes feature contours in the cooling area (Section S2.1).

The fire front delimitation is based on: (*a*) The fire front arrival time map ($t_{\text{LWIR}}^{\text{arr}}$) set with a fixed conservative temperature threshold (fire reaches a pixel when $T > T^{\text{fire}}$), and (*b*) a constant flaming residence time $t^{\text{resi}}$. The parameters $T^{\text{fire}}$ and $t^{\text{resi}}$ are adjusted

for each burn. They depend on the orthorectified image resolution and the front depth variability for the fire duration, respectively. At time $t_p$ of image $I_p$, the mask is therefore defined by pixels where $t_{\text{LWIR}}^{\text{arr}} < t_p - t^{resi}$. The feature filter applies a local normalization to a BT image to emphasize feature contours in the cooling area (Section S2.1). This filter, hereafter named local normalization and denoted by $I_p^{LN\text{rec}}$ when applied to image $I_p^{\text{rec}}$, requires as input a window size ($l^{LN}$).

As in Algorithm 1, the optimization of $H_{\text{rec}}^p$ (i.e., $H_{\text{adj}}^p$) is performed recursively with a combination of feature- and area-based alignments (see function $f_{\text{warp}}$ defined in Section S2.1) using the 10 previous images as reference templates. Algorithm 2 is summarized below.

---

**Algorithm 2** Recursive optimization of LWIR alignment

---

**for** $p = 1, N$ **do**

$\quad I_m^{\text{rec}} \leftarrow (I_0^{\text{rec}}, \rho_0), \dots, (I_{p-1}^{\text{rec}}, \rho_{p-1})$ $\qquad\qquad\qquad\qquad\qquad$ ▷ reference image update,

$\quad M_p^{\text{rec}} \leftarrow t_{\text{LWIR}}^{\text{arr}}$ $\qquad\qquad\qquad\qquad\qquad\qquad\qquad\qquad\qquad\quad$ ▷ cooling area mask

$\quad I_p^{LN\text{rec}}, I_m^{LN\text{rec}} \leftarrow LN(I_p^{\text{rec}}, I_m^{\text{rec}})$ $\qquad\qquad\qquad\qquad\quad$ ▷ cooling area feature emphasis

$\quad H_{\text{adj}}^p = f_{\text{warp}}(I_p^{LN\text{rec}}, \dots, I_{p-10}^{LN\text{rec}}, I_m^{LN\text{rec}})$ $\qquad$ ▷ alignment, see steps 4 and 5 in Figure S6

$\quad I_p^{rec} \simeq H_{\text{adj}}^p \cdot I_p^{\text{rec}}$ $\qquad\qquad\qquad\qquad\qquad\qquad\qquad\qquad\quad$ ▷ adjustment

$\quad t_{\text{LWIR}}^{\text{arr}} \leftarrow I_p^{\text{rec}}$ $\qquad\qquad\qquad\qquad\qquad\qquad\qquad\qquad\quad$ ▷ arrival time map update

---

Similarly to Algorithm 1, a steady area mask $M_p$ is defined. It is applied to the orthorectified image and fire pixel, which are filtered with a temperature threshold adjusted on a fire basis $T_{\text{Algo2}}^{\text{thresh}}$ to better suite the fire intensity. $M_p$ also removes part of the template image where previous orthorectification quality is assessed low (this can be caused by unexpected appearance of the plume or the active fire front for example). Pixels are masked out when local SSIM, averaged over the last 20 images, drops below a certain threshold $\text{SSIM}_{\text{Algo2}}^{\text{thresh}}$. This threshold also depends on the burn.

This concludes the description of the orthorectification algorithms for LWIR images with background scene. A final task of filtering outlier images within the orthorectified time series (i.e., $\text{Filter}_{\text{LWIR}}^{\text{SSIM}}$) is discussed in Section 6.1.2.

### 5.2. Algorithms for Mid Infra Red Images

The algorithms presented in Section 5.1 cannot be applied for the orthorectification of MIR images due to the limited temperature sensitivity of the Agema 550 camera [21]. This implies that fixed background features are not available to perform image alignment.

Ref. [21] designed a strategy that tracks four fixed GCPs, from image to image, to compute the homography projection. In the present work, MIR image orthorectification relies on the multi-resolution area-based alignment introduced in Algorithm 1 (see function $f_{\text{ECC}}$ defined in Section S1.2), which corresponds to a two-step algorithm. First, MIR images having near concurrent LWIR images are orthorectified using the full images, including the front and cooling areas (Algorithm 3). Second, the remaining MIR images are orthorectified using cooling area alignment with the closest processed MIR image (Algorithm 4). Concurrent LWIR images are required to observe:

$$\Delta t = |t^{MIR} - t^{LWIR}| < \left(\text{MIR frame rate}\right)^{-1} = 0.3 \text{ s.} \qquad (1)$$

Compared to [21], the new approach is more robust since it does not rely on tracking specific features, and since it has the advantage of not requiring corner fires. However, it requires one to align concurrent (MIR, LWIR) image pairs. These image pairs can show very different patterns due to the emissivity of the different fire elements (e.g., gas, soot particles, smoke), which have strong dependence on wavelength [42]. To overcome this issue, images

are filtered through the normalized gradient of the locally normalized BT, hereafter denoted by $\nabla^{LN}$. This filter helps to render image patterns consistently for LWIR and MIR. In $\nabla^{LN}$, gradient is applied to the image where pixel values are set according to local Cumulative Distribution Frequency (CDF) of BT intensity (i.e., adaptative equalization of the local normalization). BT differences due to Planck's law are therefore damped. Filtered image dissimilarity between LWIR and MIR is then mostly controlled by the local variability of emissivity. Varying emissivity can be important within the flaming front, but significantly lower in the cooling area, where it is less dependent to wavelength as it is controlled by solid material rather than hot gas or soot particles.

### 5.2.1. Algorithm 3

Algorithm 3 is summarized below. Image pairs (MIR, LWIR) are aligned on the LWIR perspective using: $(a)$ A priori warping, which corrects for the fixed perspective transformation induced by the camera orientation on the platform mount; and $(b)$ the image contours emphasized with the $\nabla^{LN}$ filter (Section S3.1). The MIR image is finally orthorectified using $H_{\text{rec}}^{lwir}$ (corresponding to $H_{\text{rec}}^{p}$ computed for the LWIR image in Algorithm 2).

---

**Algorithm 3** Orthorectification of MIR image

---

**for** $I_{lwir}$ in set of LWIR orthorectified images **do**

$\quad {}^{00}I_{mir} \leftarrow I_{lwir}$ $\qquad\qquad\qquad\qquad\qquad$ ▷ select near-concurrent MIR images

$\quad I_{mir} = {}^{00}H_{lwir}^{mir}$ $\qquad\qquad\qquad\qquad$ ▷ apply initial warp to $I_{lwir}$ perspective

$\quad I_{mir}^{\star}, I_{lwir}^{\star} \leftarrow \nabla^{LN}(I_{mir}, I_{lwir})$ $\qquad\qquad$ ▷ enhanced cooling area similarity

$\quad H_{lwir}^{mir} = f_{\text{ECC}}(I_{mir}^{\star}, I_{lwir}^{\star}, \ldots)$ $\qquad\qquad$ ▷ apply area-based alignment (see Figure S7)

$\quad I_{mir}^{rec} \simeq H_{\text{rec}}^{lwir} \cdot H_{lwir}^{mir} \cdot I_{mir}$ $\qquad\qquad\qquad$ ▷ compute orthorectification

---

### 5.2.2. Algorithm 4

To orthorectify the MIR images that were not linked with concurrent LWIR images ($\Delta t > 0.3$ s), a last algorithm, Algorithm 4, is proposed (Section S3.2). It optimizes the cooling area alignment between consecutive MIR images. MIR images processed with Algorithm 3 are available as templates at the frame rate of the LWIR sensor (1 Hz). With a 3-Hz frame rate for MIR, each unwarped MIR image is never further apart than 0.5 s from a MIR template. Over such a time window, cooling is assumed to remain unchanged and each new MIR image, $I_p$, is therefore aligned to the nearest original template perspective, $I_q$, using a cooling area mask $M_q$. This mask combines: $(i)$ A front mask set with the arrival time map of Algorithm 2 ($t_{\text{LWIR}}^{\text{arr}}$) warped on the template perspective; and $(ii)$ a fire pixel mask using a fixed MIR BT-threshold of $T_{\text{Algorithm 4}}^{\text{fire}}$, which masks out fire pixels in the fire front depth. Unlike the LWIR alignment optimization, flaming residence time is not used to mask fire pixels of MIR images. A MIR BT-threshold is preferred to maximize the cooling area, which is limited in the MIR (BT $> 470$ K). The final orthorectified image $I_p^{\text{rec}}$ is computed using $I_q$ orthorectification homography $H_{\text{rec}}^q$, which was computed in Algorithm 3.

To strengthen the orthorectification process of the whole time series, the alignment on neighbor images is coupled with a filter removing outlier images (i.e., Filter$_{\text{MIR}}^{\text{SSIM}}$, Section 6.2). The combination of alignment and filter is run twice to reduce alignment error.

---

**Algorithm 4** Iterative Optimization of MIR image Orthorectification

---

$\text{Stack}_{mir}^{rec} \leftarrow$ Algorithm 3                                               ▷ input set of MIR orthorectified images
$\text{Stack}_{mir}^{rec} \leftarrow \text{Filter}_{\text{MIR}}^{\text{SSIM}}(\text{Stack}_{mir}^{rec})$
run twice:
    **for** $I_p$ in $(\text{Stack}_{mir} - \text{Stack}_{mir}^{rec})$ **do**
        $I_q = f(\text{Stack}_{mir}^{rec})$                                                      ▷ select reference neighbor image
        $I_p^\star, I_q^\star \leftarrow \nabla^{LN}(I_p, I_q)$                                           ▷ enhanced cooling area similarity
        $H_q^p = f_{\text{ECC}}(I_p^\star, I_q^\star, \ldots)$                                             ▷ apply area-based alignment
        $I_p^{rec} \simeq H_{\text{rec}}^q \cdot H_q^p \cdot I_p \rightarrow \text{Stack}_{mir}^{rec}$                 ▷ compute orthorectification
    $\text{Stack}_{mir}^{rec} \leftarrow \text{Filter}_{\text{MIR}}^{\text{SSIM}}(\text{Stack}_{mir}^{rec})$

---

## 6. Application to KNP14 Data Set

The suite of orthorectification algorithms (Section 5) is applied to the KNP14 data set (Section 4). The algorithm parameter calibration and the image outlier filtering are discussed.

### 6.1. Application of Algorithms 1 and 2 to LWIR Images

#### 6.1.1. Algorithm Parameter Calibration

Algorithm 1 performs image orthorectification based on the background area alignment (Section 5.1). The following parameters are of first importance: Thresholds to monitor 2D correlation ($\rho_{\text{ECC}}^{\text{Ref}}$ and $\rho_{\text{ECC}}^{\text{Alg}}$) and control parameters to adjust the stack of previous images used in the alignment ($t_{\text{tail}}$, $n_{\text{tail}}$, Sections 5.1 and S1.2). Further parameters are included into Algorithm 1 to test its robustness: A switch to activate or not the area-based alignment, and a set of kernel sizes to tune mask size. Two kernels are set to expand the plot mask ($P$) and the steady area mask ($M_p$). The two kernels help to control hot pixel effects in the vicinity of burning pixels during alignment. The set of seven parameters is set up through a simple sensitivity study based on a brute force approach and discrete ranges of parameter values (see table in Figure 2). The performance of a given parameter set is evaluated through the mean value of $\text{SSIM}^{\text{prev}}$ over the last 20 images, i.e.,

$$\Gamma_1^{\text{perf}} = \frac{1}{20} \sum_{N-20}^{N} \text{SSIM}_p^{\text{prev}}.$$

This sensitivity study is applied to two fires with different fire activity: Skukuza6 and Shabeni1. Figure 2 presents $\text{SSIM}^{\text{prev}}$ time series and the parameter values that gives the best performance for both burns.

Although some of the selected parameter values belong to the range limits of the sensitivity analysis, the sensitivity analysis is not further developed since Algorithm 1 is only intended to provide a first guess of the orthorectification transformation. As explained in Section S1.1, parallax effects caused by objects in the scene (e.g., trees) and the moving camera are a critical problem when dealing with image alignment. To mitigate this effect, the methodology detailed in Section 5 imposes a near-constant line of sight between the camera and burning plot. To ease this constraint, the sensitivity analysis shows that increasing the number of images (and therefore view/template) in the stack of available images for the feature-based alignment (i.e., $n^{\text{tail}}$ varying from 1 to 4) greatly improves the performance of Algorithm 1. Increasing further the image stack has negligible improvement in regard to computational cost.

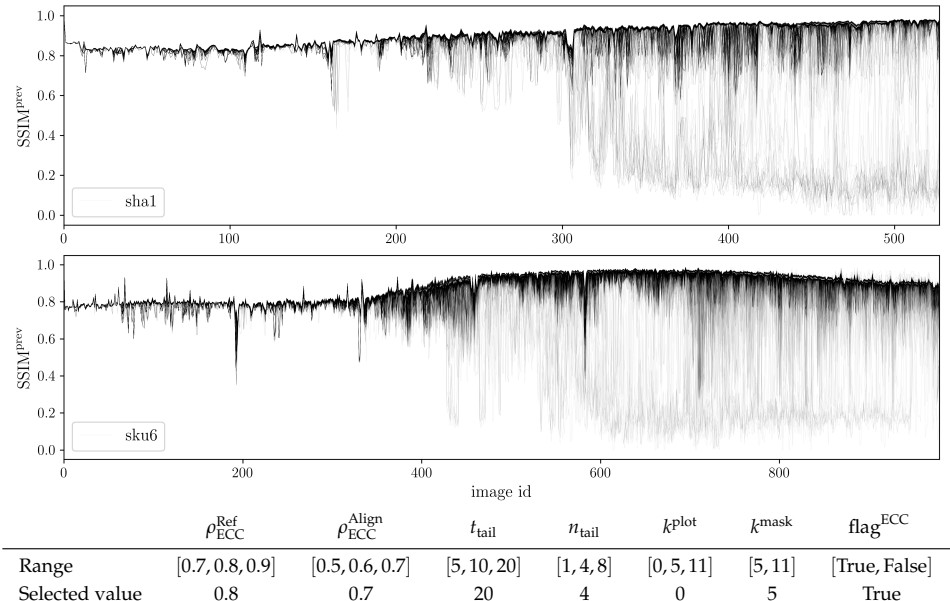

**Figure 2.** Sensitivity analysis associated with the seven input parameters of Algorithm 1. Top and middle panels show the time series of the image-to-image structural similarity index metric (SSIM$^{\text{prev}}$) for Shabeni1 (sha1) and Skukuza6 (sku6) LWIR observations, respectively. A total of 972 parameters sets are tested, and each line represents one ot them. Ideally a value of 1 is expected. Darker areas show the convergence of the calibration. Bottom panel reports the discrete ranges and the selected values for the parameters.

Algorithm 2, which recursively optimizes the alignment of the orthorectified images resulting from Algorithm 1, is based on four parameters: $T_{\text{Algo2}}^{\text{thresh}}$, $\text{SSIM}_{\text{Algo2}}^{\text{thresh}}$, $t^{\text{resi}}$, and $l^{LN}$. The values for these parameters are determined for each burn using a trial-and-error approach. Selected parameter values are reported in Section S2.2, together with explanations (e.g., data quality or fire activity) that led to this choice.

### 6.1.2. Image Outlier Filtering

To finalize LWIR image orthorectification, a filter (Filter$_{\text{LWIR}}^{\text{SSIM}}$) is applied to the image-to-image similarity measure (SSIM$^{\text{prev}}$) time series. The objective is to remove potential outliers from the images processed by Algorithm 2. It is set using rolling mean and standard deviation of SSIM$^{\text{prev}}$, removing image $I_p$ when:

$$_f\text{SSIM}_p^{prev} < \text{SSIM}_p^{prev}\Big|_{p-10:p+10}^{\text{mean}} - 2\,_f\text{SSIM}_p^{prev}\Big|_{[p-10:p-1]}^{\text{std}}, \tag{2}$$

where $_f\text{SSIM}_p^{prev}$ is calculated using images from the filtered times series. The filter is recursive as SSIM$^{\text{prev}}$ (the right-hand term in Equation (2) only depends on previous filtered images). However, to avoid a divergence towards high SSIM$^{prev}$ values, the mean in Equation (2) is computed on the initial image time series using a centered window. Figure 3 shows filter improvement to the SSIM$^{\text{prev}}$ time series when applied to Skukuza6 LWIR images.

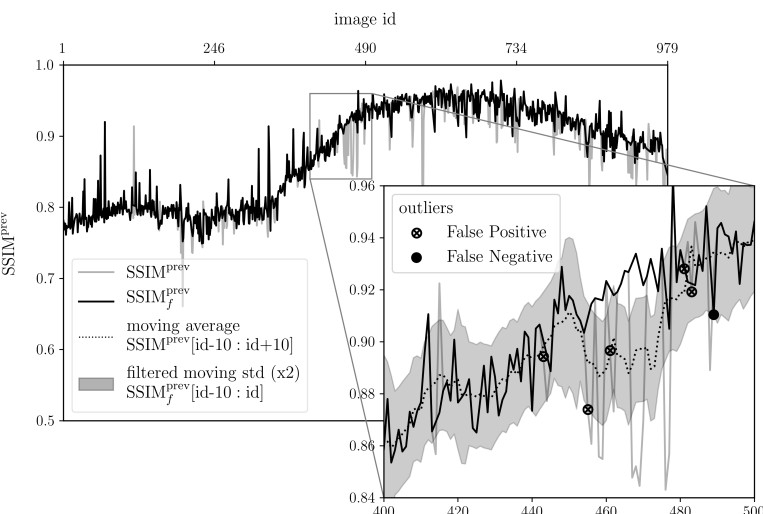

**Figure 3.** Time series of image-to-image structural similarity index metric (SSIM$^{\text{prev}}$) for LWIR observations of Skukuza6. The gray line is for the image time series output by Algorithm 2 (SSIM$^{\text{prev}}$). The black line is for the final filtered image time series ($_f$SSIM$^{\text{prev}}$). The dotted line and gray area are the centered rolling average of SSIM$^{\text{prev}}$ and the left rolling standard deviation of $_f$SSIM$^{\text{prev}}$ (Equation (2)). Encapsulated zoom shows the filter action.

The KNP14 data set relies on a theoretical camera calibration that does not correct for distortion (Section 4). This is particularly a problem for the LWIR images as the Optris camera lens has a rather poor optical quality, amplified by its large field of view (larger than the other cameras, Table 2). Image distortion is therefore penalizing both feature- and area-based alignment as it alters local pixel distribution. To improve the stability of the final orthorectified LWIR image time series, a manual check is performed. It consists of manually evaluating the alignment of the current image against its neighbors. The number of manually-flagged images, i.e., incorrectly unfiltered images (False Positive) and filtered good alignment (False Negative), accounts for less than 6% of the total initial number of images in the worst case (Skukuza6).

### 6.2. Application of Algorithms 3 and 4 to MIR Images

Algorithms 3 and 4 are designed to be less dependent on parameters than the two previous algorithms tailored for LWIR images. The only required parameters are the window size $l^{LN}$ of the local normalization used in the $\nabla^{LN}$ filter of Algorithm 3, and the fire BT-threshold $T_{\text{Algo4}}^{\text{fire}}$ used in Algorithm 4. As it is applied on the original image perspective, unlike in Algorithm 2 where it is applied to an orthorectified image, $l^{LN}$ is expressed in pixels. A constant value of $l^{LN}$ equal to 30 pixels is chosen. This value is relatively large to easily include vegetation structures that appear in the image bottom part. The lack of need to adjust for hovering altitude was noticed. $T_{\text{Algo4}}^{\text{fire}}$ is, however, tuned on a fire basis, leading to $T_{\text{Algo4}}^{\text{fire}} = 700$ K for all burns but Skukuza4 where $T_{\text{Algo4}}^{\text{fire}} = 650$ K.

Another difference from Algorithm 2 is that the image outlier filter (Filter$_{MIR}^{\text{SSIM}}$) is included in Algorithm 4 (Section S3.3). The MIR filter is based on a Hampel filter [43], which runs on a measure of low SSIM cover ratio. Figure 4 shows the application of Filter$_{MIR}^{\text{SSIM}}$ for Skukuza6. Results from the initial and final filter calls are reported. The total number of images in the MIR time series passes from 863 after Algorithm 3, to 1795 after Algorithm 4 with a time series of low SSIM cover ratio showing significantly fewer outliers.

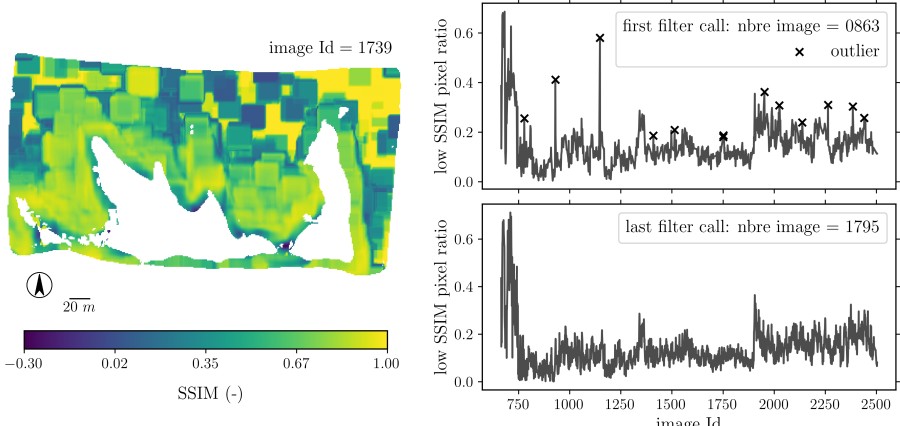

**Figure 4.** Overview of Filter$_{MIR}^{SSIM}$ used to clean the MIR image time series. Left panel shows the 20%-percentile of the SSIM similarity map for image $p = 1739$ against its 10 closest neighbor images (SSIM$_{2D\ p}^{center}$). Right panels show the time series of the cooling area cover fraction (mask $M_p$) with SSIM$_{2D\ p}^{center}$ lower than 0.2. This metric is passed through a Hampel filter to flag image outliers. Algorithm 4 is designed over a recursive loop that progressively includes MIR images unsynchronized with LWIR images. Top and down right panels show filter application at its first and last calls (Figure S8).

## 7. Discussion

This section first offers a discussion on the orthorectification accuracy of LWIR and MIR images. Then, the resulting image time series for the four burns are presented. Finally the potential exploitation to such fire observations is shown with an application to compute FRP and FRE.

### 7.1. Orthorectification Accuracy

Estimating orthorectification accuracy is necessary for the computation of several fire behavior metrics that depend on multiple images. For example, ROS which is computed from the arrival time map [44], is impacted by image registration error, in particular when the fire is slow and information from multiple images is used at a same grid point. In this case, low absolute error is required. The computation of FRE is another example sensitive to absolute registration error as it is a time integration of FRP from all images.

To estimate orthorectified image absolute accuracy, a standard method is to compare GCP true locations to those retrieved in images. For example, good features to use as GCPs would be the road/fire break present at terrain level around all the burn plots. Unfortunately, for both LWIR and MIR cameras, these features were found impossible to track. Isolated road features (without vegetation) show either with too low contrast in the LWIR images, or not at all in the MIR images. If GCPs are formed by objects above the terrain (e.g., bush, tree), the parallax effect and moving camera introduce artificial feature displacement. This displacement increases when the object is further away from the image center, and is even more amplified when camera distortion is not corrected.

Due to the lack of ground features that can be identified along the image time series, we propose here to estimate the performance of both LWIR and MIR image orthorectification using GCPs formed by the corner fires that are present in Skukuza4 and Skukuza6 burns. Despite expected artificial displacement, this approach has the advantage to be independently applicable to both LWIR and MIR images. As these corner fires are located on image edges, they are more likely to be affected by the lack of distortion correction. The methodology discussed below is therefore expected to overestimate the orthorectification error. Tests with better camera calibration will be investigated in the future.

The corner fires are made of 50-cm tall metallic cylinders filled with burning logs. In both raw LWIR and MIR images, features associated with the corner fires (base and flame/smoke) show as a vertical structure spanning over several pixels. The number

of involved pixels depends on the distance of the corner fire to the camera, view angle, and corner fire activity.

To identify the corner fires in LWIR and MIR images, we follow the same approach as in [21]. Once fire pixels are labeled with a simple threshold temperature ($T^f$) and clustered, using the initial locations that were manually selected, corner fire positions are defined recursively by: (*a*) Tracking a fire cluster with similar temperature distribution and size, and (*b*) using the temperature-weighted barycenter of the new cluster as the corner fire location.

Note that the corner fires are not included in the alignment processes of Algorithms 1, 2, and 4. They are either removed by the plot mask or a low pass temperature filter. They are only used in the first step of the MIR image alignment in Algorithm 3: Experience shows that they mostly help to start processing MIR images earlier in the time series. When corner fires are not present (i.e., Shabeni1 and Shabeni3), the first MIR image is taken later in the time series, as the algorithm requires at least two edges of the plot to be fully ignited with backfire to have enough fire pixels to start the MIR/LWIR alignment.

Figures 5 and 6 show corner fire displacement along the image time series for Skukuza6 and Skukuza4, together with the corner fire distance to original image border, and with camera poses (azimuth, tilt, and view angles). Camera poses are computed from orthorectified images, corner fire being present or not in the image, and using a priori knowledge of camera calibration. Despite the use of a priori calibration data, camera angles come to a rather good match showing noticeable biases that certainly originate from the mount where cameras were not aligned.

Top and lower panels from Figure 5 show that Skukuza6 orthorectified LWIR and MIR images do not experience drift (Section 5.1) but do have absolute displacements that fluctuate within 3 and 8 m, respectively. Up to the times $t_1^{LWIR}$ and $t_1^{MIR}$ (vertical dashed lines in Figure 5), absolute displacements are even lower, remaining within 2 and 5 m, respectively. At these specific times, the operational camera setup breaches certain of the assumptions listed in Section 5. In the case of the LWIR images, changes in the camera view angle, coupled with a corner fire getting close to the image border (where distortion is important since it is not corrected in the KNP14 data set), make one corner fire diverge from its expected location when compared to others at the same time. In the case of the MIR images, at time $t_1^{MIR}$, the upper right corner fire becomes isolated from the bulk of the active fire pixels. As Algorithm 4 is set to better adjust the part of the burning plot covered with active fire pixels, the upper right corner fire displacement is less constrained. Note that the misplacement position of the green corner fire has direct effect on the MIR camera pose estimation, which diverges from the LWIR pose. After time $t_2^{MIR}$, fronts merge and die, resulting in a more homogeneous distribution of the remaining smoldering fire pixels. Relative displacements increase slightly compared to the fire start, but the absolute displacement improves from the situation just after $t_1^{MIR}$.

Now considering Skukuza4 (orthorectified at 50-cm resolution), Figure 6 shows that corner fire displacements are similar to Skukuza6 for the LWIR images, which on average remain below 3 m. Effects from camera pose changes are noticeable. At time $t_1^{LWIR}$, a change in view angle penalizes the alignment of the corner fires located at the top of the LWIR image (red and green corner fires). At time $t_2^{LWIR}$, a change in azimuth angle produces a drift of the red corner fire, which is at that time located the furthest away from the camera (i.e., with the poorest resolution). Meanwhile, the two corner fires located at the bottom of the image remain within a distance of 2 m from their initial position. Displacement of corner fires in the MIR images are, however, much smaller, remaining below 4 m over the fire duration. Until time $t_1^{MIR}$ when one corner fire (green corner fire) gets close to the image borders and even disappears from the field of view (see black background on third panel from bottom), displacements are even better, remaining around 2 m. After the four corner fires get back in the field of view, relative displacements are not so much impacted; only small drift (less than 2 m) shows on the upper corner fire (red and green corner fires). Unlike for Skukuza6, the MIR camera for Skukuza4 is not equipped with a filter, hence

enabling lower temperature detection. This feature coupled with the higher data resolution makes the cooling area much better resolved in MIR images. This is also certainly improved by the larger fuel load, which increases the smoldering time compared to Skukuza6.

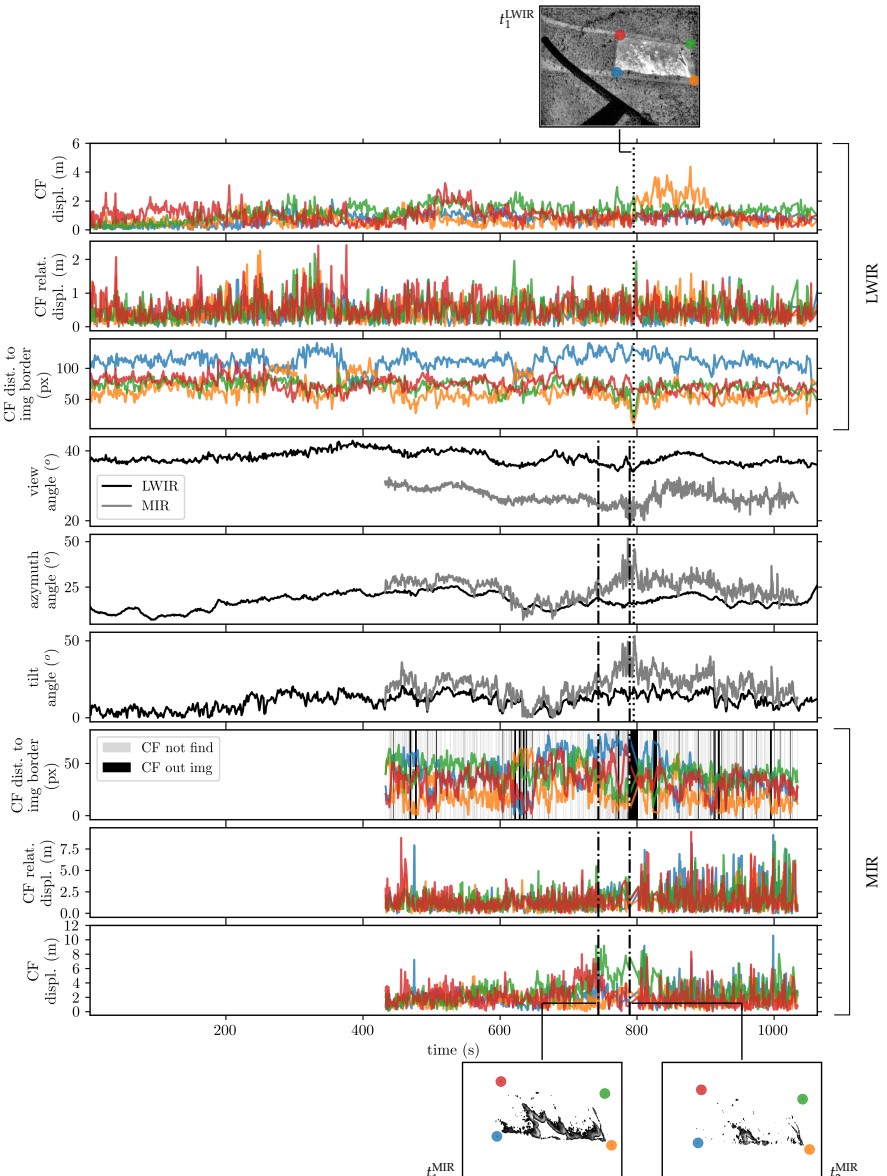

**Figure 5.** Overview of LWIR and MIR images orthorectification accuracy for Skukuza6. The three top panels show the LWIR camera time series of corner fire (i.e., GCPs) displacements from their initial positions in the orthorectified image, the same corner fire displacement but relative to previous image, and corner fire distance (in pixels) to the image edge. The three middle panels show the evolution of LWIR and MIR cameras angles (namely the view, azimuth, and tilt angles). The camera pose is computed from the image alignment using a priori geometrical calibration camera data. The three bottom panels are the same as the top three panels but for the MIR camera. Gray and black vertical lines (in seventh panel from top) indicate times where at least one corner fire is missed by the tracking algorithm, and times where at least one corner fire is out of the field of view. Encapsulated images at the top and bottom show, within the raw image, the corner fire location, and the fire stage at specific times.

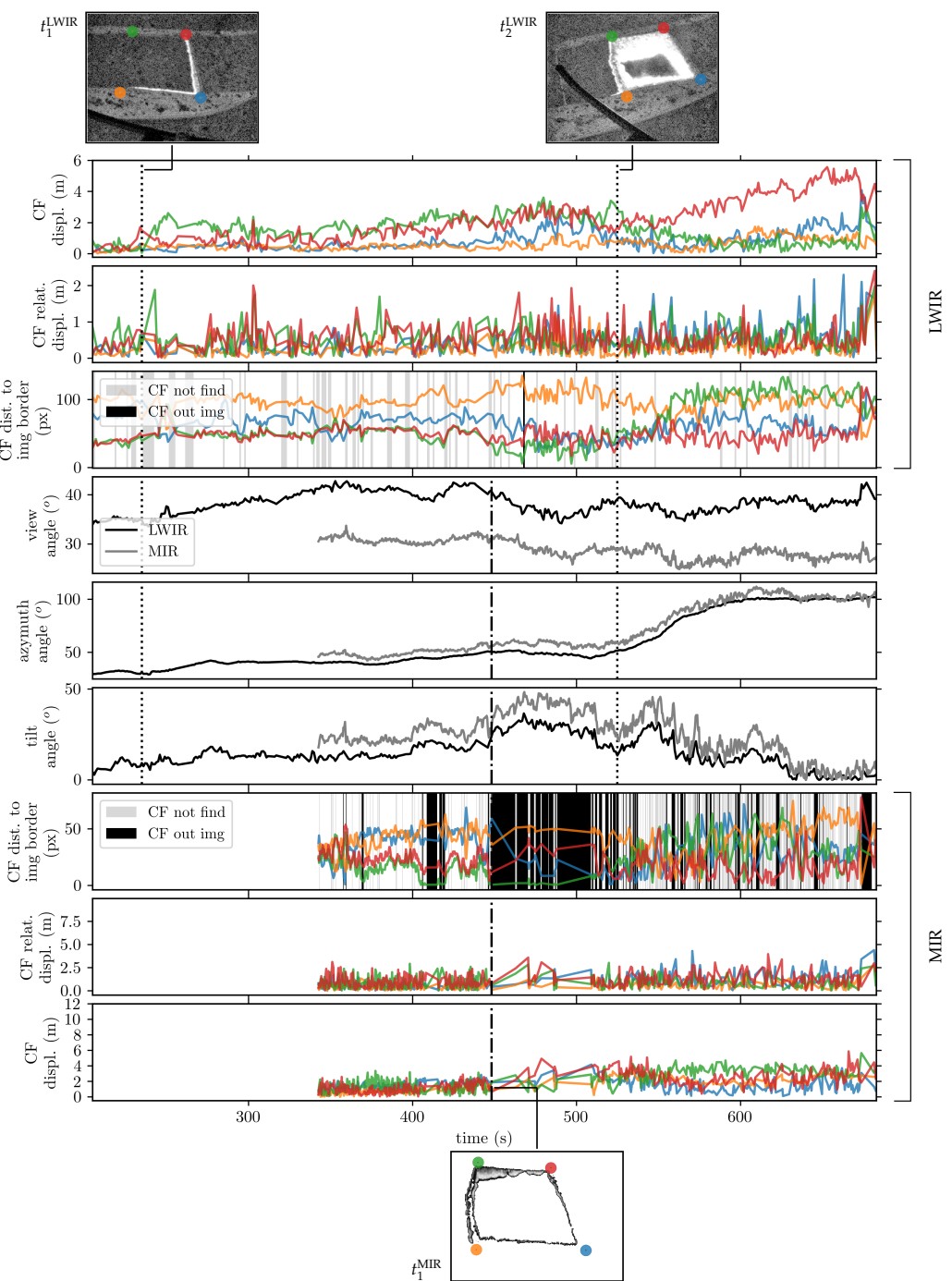

**Figure 6.** Same as Figure 5 for Skukuza4.

These results show that the alignment procedure of LWIR and MIR images performs better when feature appearance in the images is conserved. When changes are introduced (either induced by camera pose and/or distortion), features end up looking different from what the algorithms know from the available template images, and alignment quality is degraded. In the case of the MIR images (which do not show background), this is even more evident. Results from Skukuza4 show the importance of being able to capture the cooling area to improve MIR image orthorectification. To this end, the resolution increase (50 cm) helps improving the performance of Algorithms 3 and 4. However, the same resolution increase in LWIR does not provide the same improvement.

Both Skukuza6 and Skukuza4 burns show relative displacement in LWIR images of about 1.5 m and absolute displacement of about 3 m. For Skukuza6, if we do not consider times after $t_1^{MIR}$ that are impacted by the poorly resolved cooling area, MIR images show at the same time relative and absolute corner fire displacements of 2.5 and 4 m, respectively. This misregistration is of the same order of magnitude as artificial displacement (parallax effects) from tall shrubs and trees that can potentially occur during the alignment of the background (Algorithm 1) and of the cooling areas (Algorithms 2–4). Similar accuracy of the LWIR orthorectified images from Skukuza4 and Skukuza6 burns shows that we certainly reach the limit of the single vantage point, and that stereovision would have to be considered to get better alignment [45].

As discussed earlier, fire front misregistration is important when dealing with fire behavior metrics computation. Two sources of misregistration can be reported: Orthorectification, and segmentation. On the one hand, our methodology provides orthorectified images with an accuracy of at least 4 m (which can be lowered to 1.5 when only dealing with neighbor LWIR images). On the other hand, new fire front segmentation methods [46] report errors from 2 to 10 m on similar off-nadir experimental burn observations. Segmentation in orthorectified images obtained from homography transformation is affected by the flame vertical structure, which is distorted onto the projection gridded plan. With flames certainly higher than 4 m in the active front area and view angle of about 40° (e.g., Figure 5), segmentation error could be potentially larger than orthorectification error. For fire behavior metrics computation, this error accumulation will need to be addressed to make the estimation of high-resolution fire behavior metrics possible. Nevertheless, it is worth noting that the final product delivered by the present four algorithms shows a good robustness to fire activity. In the four image time series considered here, the fire activity peak never impacts orthorectification. This is a strong point of the algorithms since complex fire behavior usually occurs with strong active fire.

### 7.2. Resulting KNP14 Data Set

The final product delivered for each burn by the algorithm suite (Section 5) is formed by the combination of the orthorectified image time series delivered by the three cameras: LWIR, MIR, and VIS. Figure 7 shows overlay of LWIR and MIR contours over VIS images for Skukuza6. The final frequency of image time series (after orthorectification and outlier filtering) is on average for LWIR images between 0.3 and 2 Hz with large fluctuation among burns, and for MIR images between 1.5 and 3 Hz. Several gaps up to 10 s exist in both the LWIR and MIR time series, but never represent more than 5% of the images in one burn. In the worst scenario, 13.5% of raw images were disregarded (MIR images from Shabeni3). A dedicated web page to the KNP14 data set (https://3dfirelab.eu/knp14, last access date: 24 November 2021) proposes an overview of the full data set and an interactive display of the orthorectified images. The VIS images were orthorectified using Algorithm 1 and manually-tuned parameters. VIS images overlay with concurrent LWIR and MIR images features good results while the plume does not cover too much of the image, and/or is not too opaque. As shown on the dedicated web page, the VIS camera (from which the IR filter was removed) shows good ability to detect the flaming area, even in relatively dense smoke. A smoke mask is however necessary to improve the robustness of VIS orthorectification.

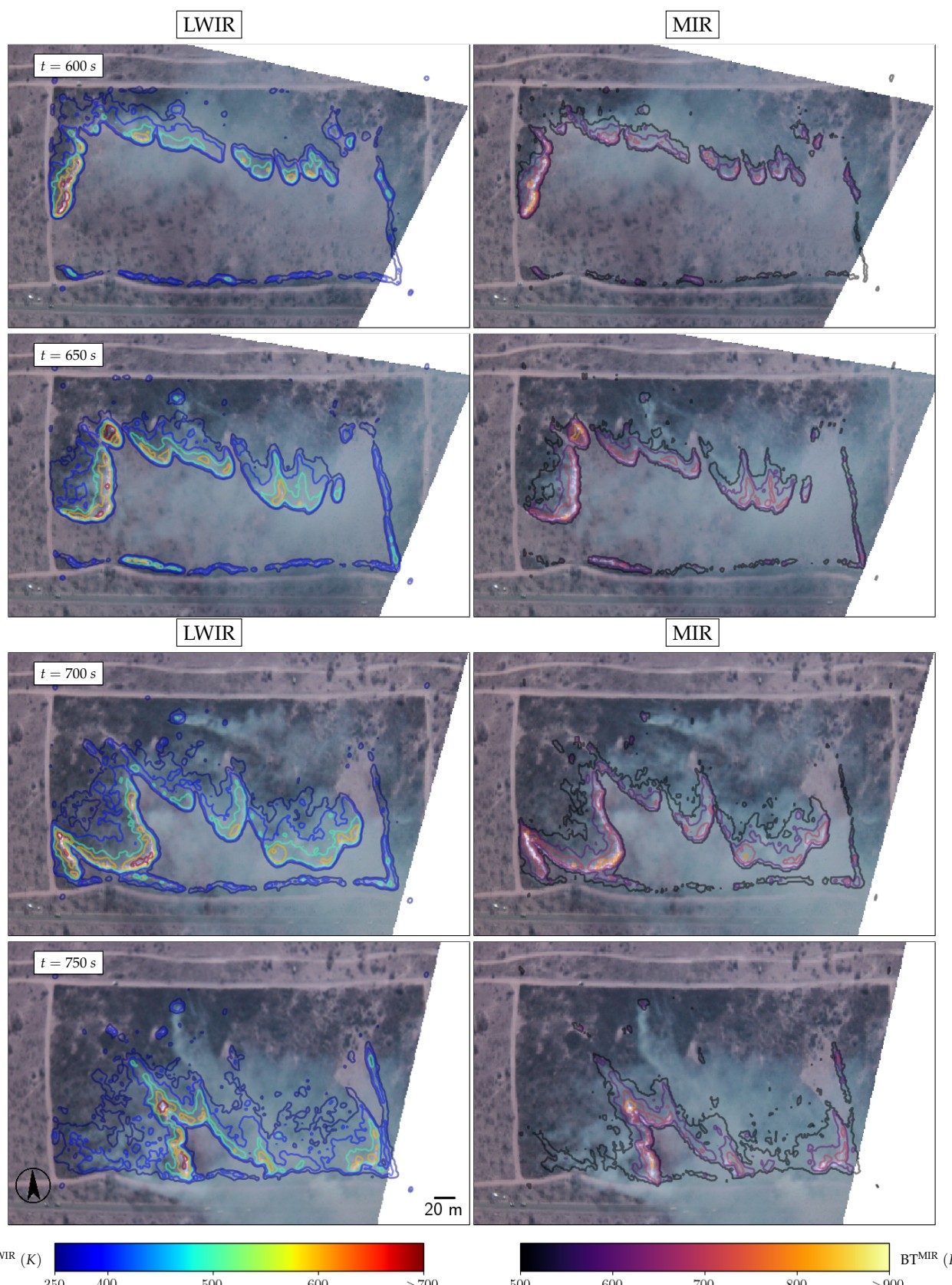

**Figure 7.** LWIR (left panels) and MIR (right panels) Brightness Temperature (BT) contours overlaid over VIS images for Skukuza6 around the time of the peak fire activity. Contours are extracted from orthorectified images processed with the algorithm suite (Section 5). All maps have the same scale and orientation as reported on the bottom LWIR image.

### 7.3. Application to Fire Radiative Power Time Series Estimation

To compute metrics associated with front dynamics (e.g., ROS, front depth), fire front segmentation is required (i.e., step *b* introduced in Section 2). However, we can already compute spatially-integrated metrics like FRP from the MIR image time series using the formulation of [47], in order to provide insight to burns fire activity.

The FRP emitted by a pixel is then defined by:

$$\text{FRP}_{px} = \frac{\sigma}{a\,\tau_{MIR}}\,s_{px}\,(L_{px}^{f} - L_{px}^{b}) \tag{3}$$

where $\sigma$ is the Stefan–Boltzmann constant, $a$ is a sensor-specific constant, $\tau_{MIR}$ is the atmospheric transmittance between the fire and the sensor, $s_{px}$ is the pixel size, and $L_{px}^{f}$ and $L_{px}^{b}$ are the MIR radiance emitted by a pixel containing fire (fire pixel) and its associated background radiance. Assuming that, at the resolution of our observation ($<1$ m) the fire spatial distribution is fully resolved ($L_{px}^{f} \gg L_{px}^{b}$), and that at the altitude of the helicopter the atmosphere is transparent in the MIR spectral region, Equation (3) simplifies to:

$$\text{FRP}_{px} = \frac{\sigma}{a}\,s_{px}\,L_{px}^{f}. \tag{4}$$

As the FRP is a conservative quantity, the perspective warping applied to raw MIR images based on the homography matrix computed in Algorithm 4 is modified to ensure energy conservation. Instead of linearly interpolating radiance values at every mesh point of our fixed grid from the un-gridded points provided by the homography transformation (i.e., Equation (S2) in Section S1.1), a perspective transformation applied at a pixel level is developed. The objective is to split each warped pixel from the original MIR image on the mesh of the assumed flat terrain, thus providing a temporal map of $\text{FRP}(x,y,t)$. This is much more time consuming and is therefore only performed once using the final optimized homography transformation of Algorithm 4.

For each burn plot, Figure 8 shows the FRP time series over the experimental burn duration, the time-integrated FRP map (i.e., the FRE), and the MIR image overlaid over the VIS image at the time of the satellite overpass that was targeted for each burn (Table 1). FRP peak values range from 120 MW for Skukuza6 to 800 MW for Shabeni3. FRP error is also reported in Figure 8. For now, only geometrical effects are considered (see pixel size term in Equation (4)). The low number of saturated pixel counting at a maximum 0.1 % of fire active pixels (See Shabeni3 in Figure 8) is neglected. Increase by 50% of the saturated pixel radiance only showed marginal FRP changes. The high radiance sensibility of the Agema 550 is also neglected ($\text{BT}^{\text{MIR}} > 470$ K). This threshold results in an underestimation of the FRP that depends on the fire activity. In the case of the Skukuza6 burn where the camera was operated with a filter ($\text{BT}^{\text{MIR}} > 520$ K), the underestimation is even higher. In a future work, the integration of LWIR data in the calculation of FRP in the cooling trail of the fire will be considered.

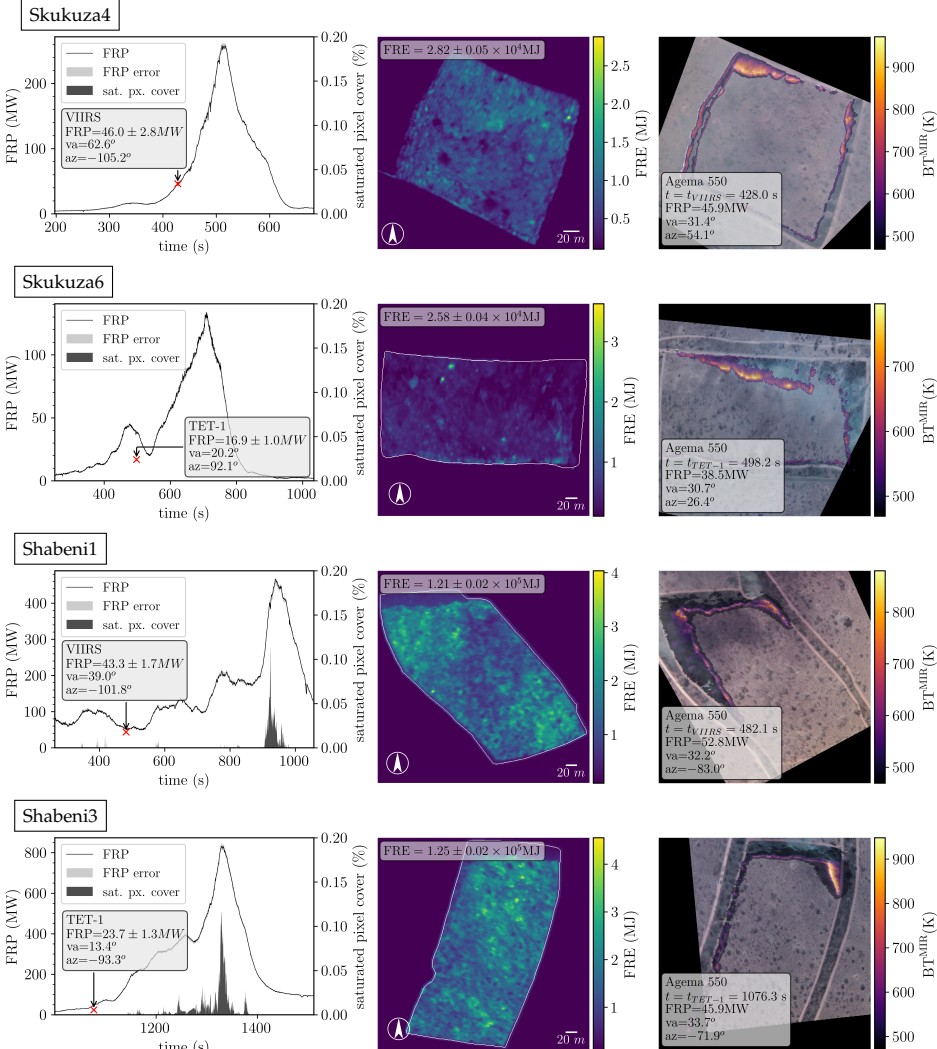

**Figure 8.** Fire Radiative Power (FRP) time series with associated MIR saturated pixel fraction (left panel) and Fire Radiative Energy (FRE) map (center panel) for the KNP14 four experimental burns. Information from the FRP product of concurrent satellite overpass are reported on the time series (va and az are the view and azimuth angles). See the text for details on satellite FRP error bar estimation. Right panels show helicopter-borne MIR and VIS images at the time of the satellite overpass.

Assuming that all burns are observed from similar camera poses, a map of pixel size error is computed using all available MIR images from Skukuza4 and Skukuza6 burn. For every pixel of every raw images, pixel size relative difference between the orthorectified standard output of the algorithms suite and the orthorectified output corrected to match corner fire displacements is calculated. The 10th and 90th percentiles of this relative error at every pixel of the raw image frame are used to estimate error bounds. The relative pixel size errors hence ranges from $-6$ to 10% with a spatial average for the lower and upper bounds equal to $-2$ and 2%, respectively. When applied to the FRP calculation of the 4 burns, this results in geometrical resampling error associated the FRP estimation that never goes above 8%.

Note that to complete the FRP error estimation, as the fire scene is not a Lambertian emitter (e.g., plume absorption, flame tilt), the effect of the camera pose on the measure of the radiance would also have to be estimated. This is however beyond the scope of the present work.

As stated in [48,49], the use of high-resolution IR fire observations would greatly contribute to revisit the current assumptions made in fire emission estimates based on FRP measurements. FRP-based emission models used in the atmospheric model e.g., [50] were

partly designed on small-scale fire measurements [47], and not enough data are currently available to validate the upscaling impact [15,33,51]. With only four burns, the KNP14 data set is too small to derive a robust statistical conclusion. However, it provides insight into two questions relevant to the use of the FRP product for fire emission estimation: (*a*) The relationship between FRE and Fuel Consumption (FC) (Figure 9—left panel), and (*b*) the FRP computation upscaling from helicopter- to satellite-borne observations (Figure 9—right panel).

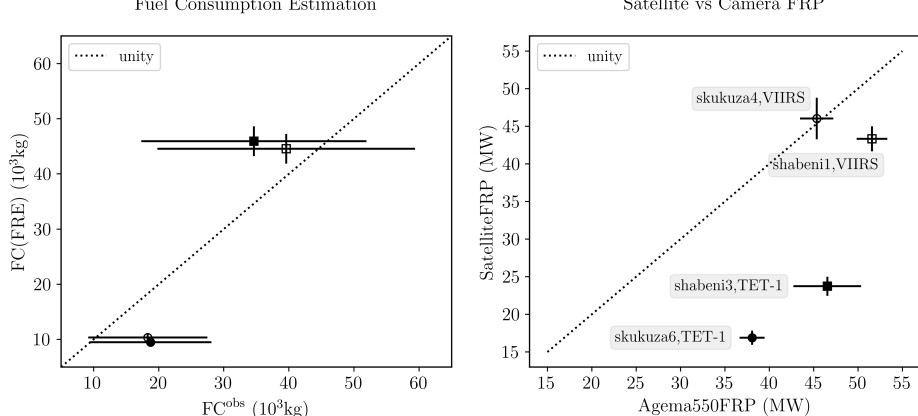

**Figure 9.** (**Left panel**): Comparison between Fuel Consumption (FC) from in-situ measurement (FC$^{obs}$, see Table 1) and estimated from Fire Radiative Energy (FC(FRE)) using the relationship of [47] for the four experimental burns of the KNP14 data set. (**Right panel**): Fire Radiative Power (FRP) comparison between satellite- and helicopter-borne images for the same four burns. Markers are identified to burn in the right panel.

FC is calculated assuming a full combustion of grass, whose fuel load was measured using a limited number of pre-fire destructive samples, which were essentially performed on the plot edges. As the fuel load measures (FC$^{obs}$) do not take into account the fuel spatial distribution across the burn plot (see for example north-eastern area of Skukuza6 in the visible image presented in Table 1), FC$^{obs}$ is therefore associated with a large error estimated here to at least 50 %. More fires with better FC measures are required to give any conclusion. Despite this large error and the underestimation of FRE inherent to the Agema 550 temperature threshold, the comparison between FC$^{obs}$ and FC(FRE) derived from the small-scale experimental estimation of [47] shows good agreement (see Figure 9—left panel).

The space-borne FRP of the KNP14 data set are computed using the MIR formulation of [47] (Equation (3)) applied to the 178-m TET-1 and the 750-m VIIRS data. The 375-m VIIRS data were not included in the analysis as the sensor saturated for the two burns it overpasses (Skukuza4 and Shabeni1). For our 4 overpasses, in order to limit error from the estimation of the background radiance ($L^b$ in Equation (3)), the fire pixel mask and $L^b$ were manually estimated. Therefore the source of surface FRP uncertainty is essentially controlled by the transmittance error. The error bars of the satellite FRP values in Figure 9 are computed using transmittance ($\tau^{MIR}$) associated to a different atmospheric profile. The Modtranv5 radiative transfer model [52] is used to compute ($\tau^{MIR}$) and atmospheric profiles representative of the day of the burn are set (*a*) varying ambient $CO_2$ concentration from 360 to 420 ppm and (*b*) setting the water vapor profile to either the standard mid-latitude summer profile or profile extracted from the European Centre for Medium-range Weather Forecasts analysis data. The error bars reported for the Agema 550 FRP values in Figure 9 are estimated based on the geometrical error mentioned above plus a potential time misregistration of 3 s between the Agema and satellites. KNP14 cameras were manually time synchronized using a handheld GPS unit. A time shift of 3 s is a conservative error range. The comparison helicopter-satellite-borne FRP (Figure 9—right panel) show good agreement between the VIIRS sensor and Agema 550, even in the case of Skukuza4, which

comes with a large VIIRS view angle (>60°). The TET-1 sensor however shows a clear underestimation of FRP by almost a factor 2. This is not in agreement with the work of [53] which reported a 31% concordance between TET-1 and VIIRS using the same FRP formulation and synchronized overpasses of the same fire. As the TET-1 FRP seems to scale properly between Skukuza6 and Shabeni3 FRP, this does not seem to be a saturation problem. The KNP14 fires took place in the early days of the TET-1 operation, a bias might has been present and corrected later. The oldest fire in [53] is from 2016. Using only the four fires from the KNP14 data set, it is difficult to draw any conclusion on the upscaling of FRP measurement. More concurrent observations with more various satellite view angle would be necessary. The methodology presented here however shows that it can be used to provide FRP calulation from sevral hectares burn with high geometrical precision leading to a FRP error lower than 8%.

## 8. Conclusions

In this work, we presented a methodology to orthorectify helicopter-borne observations from savannah experimental burns. A suite of algorithms was designed to map high-resolution radiance measurements collected with handheld LWIR and MIR thermal cameras. It was successfully applied to four burn plots ranging in area from four to eight hectares, resulting in explicit maps at a spatial resolution of 1 m (50 cm in the case of the smallest burn plot) and at a frequency close to the imager frequency acquisition. Orthorectification accuracy is estimated to be within the range of errors associated with parallax effect induced by the moving camera and flickering flames. The main requirements of the methodology are the following: The camera points towards the plot along a near-constant direction (no spinning around the plot), the background scene around the plot is kept as much as possible in the field of view (no zoom inside the plot area), and the plot can be approximated as a planar surface. If these restrictions are satisfied, it is then possible to orthorectify images without the presence of fixed ground control points during the whole fire duration. The iterative structure of the algorithm ensures the alignment to the first image, which is manually orthorectified.

The present methodology offers a way to map fire behavior, such as an energy-released map or front merging at a scale and a level of detail that is not present in the literature yet. This provides data that can clearly contribute to the current open questions in fire emission and fire model development efforts. In this sense, a comparison between helicopter- and space-borne fire observation points out the need of larger data set to better scan satellite observation angle ranges and better validate the upscaling of FRP measurement.

Collecting detailed information of fire front propagation would also contribute to the further development of data assimilation for application in fire growth modeling [54,55]. The impact of the assimilation time interval on the data assimilation performance is currently an open question, which requires access to data with very high temporal resolution. Application of the methodology to new fire scenarios is being undertaken and with ease thanks to the algorithm parameters definitions presented in this work and the experience gained on the KNP14 data set. In particular, application to fire observations collected from a UAV quadracopter is investigated. Using such a hovering platform would allow a lower view angle and would increase flight/camera pose stability, providing improvements to the algorithm performance. Another potential development is the improvement of the visible image orthorectification, which requires a plume/smoke mask to run through optimization similar to the LWIR images of Algorithm 2. A visible camera with a removed IR filter has potential to map ROS and flame depth from fires with weak plume at a very low cost.

**Supplementary Materials:** The following are available online at https://www.mdpi.com/article/10.3390/rs13234913/s1.

**Author Contributions:** Conceptualization, R.P.; Data curation, G.R., O.F., E.L., W.S., B.M. and N.G.; Formal analysis, R.P. and J.-B.F.; Methodology, R.P.; Resources, M.J.W., W.E.M., M.C.R. and J.-B.F.; Supervision, M.J.W. and W.E.M.; Visualization, R.P.; Writing—original draft, R.P.; Writing—review & editing, M.J.W., W.E.M. and M.C.R. All authors have read and agreed to the published version of the manuscript.

**Funding:** This work has received funding from the European Union's Horizon 2020 research and innovation program under the Marie Skłodowska-Curie, grant H2020-MSCA-IF-2019-892463. Parts of this work were also supported by the NERC, grant NE/M017729/1 made by the UK's Natural Environment Research Council. The KNP14 Fieldwork campaign was also supported by the Ministry of Economy of Bavaria (Az. 20-08-3410.2-05-2012) and a START grant (www.start.org, last access date: 24 November 2021).

**Acknowledgments:** The authors would like to dedicate this paper to the memory of our co-author Eckehard Lorenz, principal investigator of the FireBird mission, who sadly passed away before the manuscript could be completed. The authors would also like to thank Anja Hoffman, the South African National Parks (SANParks) and the fire team from Kruger National Park for the organization of the experimental burns as well as the various staff and students of the Department of Geography, King's College London, and other visitors, who graciously assisted with the different aspects of the fieldwork. Thanks also to the Bibliothèque Francois Mitterrand in Paris for opening its door to academics forced to pause.

**Conflicts of Interest:** The authors declare no conflict of interest.

## Abbreviations

The following abbreviations are used in this manuscript:

| | |
|---|---|
| IR | Infra Red |
| BT | Brightness Temperature |
| LWIR | Long Wave Infra Red |
| MIR | Middle Infra Red |
| VIS | Visible |
| GCP | Ground Control Point |
| SSIM | Structural Similarity Index Metric |
| ROS | Rate Of Spread |
| FI | Fire Intensity |
| FRP | Fire Radiative Power |
| FRE | Fire Radiative Energy |

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
