# Peer review of "Orthorectification of Helicopter-Borne High Resolution Experimental Burn Observation from Infra Red Handheld Imagers"

_remotesensing, doi:10.3390/rs13234913_

Round 1
Reviewer 1 Report
Not sure, what the authors mean by orthorectification. Do they mean horizontal georeferencing/georectification? Orthorectification is mostly used for correcting the effects of topography.
Acronyms are mentioned at the end of the MS but they should be mentioned in the text whenever they are mentioned for the first time.
The introduction needs to be rewritten
There are many incorrect statements in the MS, following are few examples:
Page3:
Both Long Wave Infra-Red [LWIR, 17,18] and Mid Infra-Red [MIR, 19] imagers have been used to collect IR images from burns of several hectares at high spatial (metric) and temporal (> 1 seconds). resolution. Why spatial resolution is not mentioned.
Hovering Unmanned Aerial Vehicle (UAV) platforms are also quite promising for prescribed/experimental burn applications. However, their use is still restricted to small burns [10 m
…orthorectification, which consists in warping the raw images on a fixed coordinate system grid to correct for camera lens distortion and perspective effects induced by camera orientation and terrain….
Are authors using terrain correction? Are they using a DEM?
What is the meaning of ‘landscape-scale >100m? Are authors referring to map-scale? If so, this has to be related properly to map-scale and GSM.
Space-based remote sensing images used for comparison are too coarse.
In the Background section, the authors are mentioning that globally fire events are decreasing which is not true under changing climatic conditions.
Also, there are several remote sensing data products available on flammability and past burns.
The initial part of the background section talks about the fire spread model while the title is orthorectification.
IR is a wide spectrum and NIR, SWIR and thermal IR are used for different purposes. Which spectrum is used in this study? The author must clarify this in the beginning.
All geographic informatiion presented should be of cartographic quality.
Reviewer 2 Report
I reviewed the paper in detail, in which the problem definition, background, methodologic design and experiments are given in a very solid and detailed way. The supplementary document is also prepared in an informative way. There are few writing and punctuation errors that can be easily solved by a recheck of all documents.
In Introduction Authors informed that when modelling the fire front dynamics (especially the modelling of the fire spread and determining the spatial distribution of fire within time) benefits from remotely sensed data. It is known by the audience that generally observations are collected by a plane based acquisition (Airborne systems), but an IR observation from a helicopter platform may increase the temporal resolution and improve the efficiency of modelling as stated by authors. However, collecting the IR from such platform design brings a disadvantage as acquisition geometry is not stable and mathematical geometric correction methods require ground control points.
At this point, I evaluate that Authors proposed a significant series of approaches first to handle the geometric correction procedure and then to retrieve the modelling parameters.
The novelty lies behind the removal of GCP need it orthorectification and a stabilized image to image registration where sensor model is not available. I think enough credit is given to previous works and their limitations in terms of geometric correction in the Section 5.
There are enough number of experimental sites to validate their approach. Although accuracy evaluation is different from what is usual, this way of graph based visualization is better to understand as there are so many data collection (high temporal resolution). Moreover, abstract accuracy results are also explained in dedicated discussion section (7.1).
The FRP data production and mapping is also presented in the study and experimental study is concluded with higher geometric accuracy results in better FRP results.
Conclusion summarizes the findings well and drives a certain future research aspect.
Reviewer 3 Report
The work is very interestingmbecause it improves and facilitates the orthorectification process of the images taken by UAV, for example even without the definition of GCP with a very low error compared to the same images taken by satellite.Author Response
please see attachment
